# Stable structures or PABP1 loading protects cellular and viral RNAs against ISG20-mediated decay

Camille Louvat[1],*, Séverine Deymier[2],*, Xuan-Nhi Nguyen[2], Emmanuel Labaronne[3], Kodie Noy[2,4], Marie Cariou[2], Antoine Corbin[2], Mathieu Mateo[2,4], Emiliano P Ricci[3], Francesca Fiorini[1],†, Andrea Cimarelli[2],†

ISG20 is an IFN-induced 3′–5′ RNA exonuclease that acts as a broad antiviral factor. At present, the features that expose RNA to ISG20 remain unclear, although recent studies have pointed to the modulatory role of epitranscriptomic modifications in the susceptibility of target RNAs to ISG20. These findings raise the question as to how cellular RNAs, on which these modifications are abundant, cope with ISG20. To obtain an unbiased perspective on this topic, we used RNA-seq and biochemical assays to identify elements that regulate the behavior of RNAs against ISG20. RNA-seq analyses not only indicate a general preservation of the cell transcriptome, but they also highlight a small, but detectable, decrease in the levels of histone mRNAs. Contrarily to all other cellular ones, histone mRNAs are non-polyadenylated and possess a short stem–loop at their 3′ end, prompting us to examine the relationship between these features and ISG20 degradation. The results we have obtained indicate that poly(A)-binding protein loading on the RNA 3′ tail provides a primal protection against ISG20, easily explaining the overall protection of cellular mRNAs observed by RNA-seq. Terminal stem–loop RNA structures have been associated with ISG20 protection before. Here, we re-examined this question and found that the balance between resistance and susceptibility to ISG20 depends on their thermodynamic stability. These results shed new light on the complex interplay that regulates the susceptibility of different classes of viruses against ISG20.

## Introduction

The IFN-stimulated gene of 20 kD (ISG20) is an antiviral RNase that belongs to the DnaQ-like exonuclease superfamily, which is highly conserved across prokaryotes and eukaryotes. Members of this superfamily present a common catalytic core defined by four conserved aspartate and glutamate residues (DEDD) (1, 2). Among the members of this family, ISG20 was identified in 1997 as a type I IFN–induced protein and was soon associated with RNA virus inhibition (3, 4, 5, 6). Over the years, the spectrum of viruses described to be inhibited by ISG20 has expanded to encompass several positive- and negative-strand RNA viral families, in addition to retroviruses (7, 8, 9, 10, 11, 12, 13, 14, 15, 16, 17, 18, 19, 20, 21). However, a number of studies have also identified viruses that exhibit a strong resistance to this antiviral factor, pointing on the whole to the existence of a likely complex relationship between ISG20 and viruses that remains to be unraveled (11, 16, 17).

Similarly, the exact mechanism of viral inhibition by ISG20 remains unclear (6). In light of its strong RNA exonuclease activity, the first mechanism of viral inhibition proposed for ISG20 was based on the direct degradation of viral RNA (4, 5). This model was supported by the fact that ISG20 behaves as a potent RNase in vitro and that lower levels of viral RNA were often, albeit not always, measured in ISG20-positive cells undergoing infection. Also in agreement with this model, point mutations in the catalytic core of ISG20 compromised both its ability to degrade RNA in vitro and its ability to inhibit viral replication in cells (4, 5). However, several studies failed to report a strong decrease of viral RNA levels despite clearly measurable antiviral effects of ISG20, raising the possibility that alternative mechanisms, among which is translation inhibition from viral RNAs, could be at play (11, 16, 17, 18).

A recent feature that gained interest as a possible explanation for how ISG20 could discriminate between RNAs has been the presence of epitranscriptomic modifications, such as $N^6$-methyladenosine ($m^6A$) or 2′O-methylation (2′O-Me) (19, 22). These and other modifications are added on cellular RNAs and can influence virtually every aspect of their metabolism, among which are nuclear export, stability, or translation rates. Furthermore, RNA $m^6A$

[1]Molecular Microbiology and Structural Biochemistry, MMSB-IBCP, UMR 5086 CNRS University of Lyon, Lyon, France  [2]Centre International de Recherche en Infectiologie (CIRI), Université de Lyon, Inserm, U1111, Université Claude Bernard Lyon 1, CNRS, UMR5308, École Nationale Supérieure de Lyon, Lyon, France  [3]Laboratoire de Biologie et Modelisation de la Cellule, Université de Lyon, ENS de Lyon, Université Claude Bernard, CNRS UMR 5239, Inserm, U1293, Lyon, France  [4]Unité de Biologie des Infections Virales Emergentes, Institut Pasteur, Lyon, France

Correspondence: francesca.fiorini@ibcp.fr; acimarel@ens-lyon.fr
Marie Cariou's present address is Service Analyse de Données, Muséum National d'Histoire Naturelle, Centre National de la Recherche Scientifique, Paris Cedex, France
*Camille Louvat and Séverine Deymier are Joint first authors
†Francesca Fiorini and Andrea Cimarelli are Joint last authors

modifications are dynamic, as they can be removed by dedicated enzymes referred to as erasers and can also act as docking sites for specific proteins called readers, overall accounting for the pleio-tropic influence that epitranscriptomic modifications play on the behavior of RNAs (reviewed in reference 23). An increasing number of studies are depicting a complex canvas on how viruses them-selves can co-opt such modifications for their own purposes, among which is to mimic cellular RNAs (reviewed in reference 24). In this respect, the presence of a single m6A modification at a specific location on the hepatitis B virus viral RNA has been recently de-scribed to promote ISG20-mediated RNA degradation, through the recruitment of a m6A reader protein YTH N$^6$-methyladenosine RNA-binding protein F2 (YTHDF2)–ISG20 complex (19). However, an opposite behavior has been described in the case of 2'O-Me modifications that are scattered along the human immunodefi-ciency type 1 virus (HIV-1) ≈10-kilobase genome and that act as protective elements against ISG20-mediated degradation (22). The fact that epitranscriptomic modifications can modulate the sus-ceptibility of RNAs to the action of ISG20 opens up the unaddressed question of how this enzyme behaves toward cellular RNAs in which these modifications are largely present. Lastly, RNAs are often associated with proteins that also influence their biology and it is unclear how association with specific viral or cellular proteins may modulate the susceptibility of target RNAs to ISG20. For example, the poly(A)-binding protein (PABP1) is the best studied cellular protein that associates with the poly(A) tails present in all cellular mRNAs with the exception of histone mRNAs and influences several aspects of their biology (translation, stability, etc.) (25).

To more globally appreciate the effects of ISG20 on RNAs at the scale of the whole cells, we performed an RNA-seq analysis on cellular mRNAs. The results we have obtained indicate that ISG20 expression does not lead to overt changes in the cellular tran-scriptome, suggesting that cellular mRNAs are generally protected against ISG20-mediated degradation. However, a small effect was observed on histone mRNAs that represented the main category of mRNAs down-regulated in the presence of ISG20. This finding is of interest because histone mRNAs are the only mRNA species in the cell devoid of poly(A) tails, but instead exhibit a 3' stem–loop structure (26), which is surprisingly similar to those that are often described as protective against ISG20-mediated degradation (14). This prompted us to re-examine the reasons and the determinants that govern the resistance of poly(A) cellular mRNAs on the one hand and the susceptibility of stem–loop-bearing RNAs using finer biochemical analyses that used the purified recombinant ISG20 protein and synthetic model RNAs on the other. The results we have obtained indicate that PABP1 provides a key layer of protection against ISG20, by shielding the 3' extremity of cellular mRNAs, on the one hand providing the explanation for the resistance of cellular mRNAs from ISG20 and on the other opening the question of the interplay between ISG20 and the plethora of viral and cellular binding factors that often decorate RNAs. In the case of non-poly(A) mRNAs, we confirm that stem–loop structures can similarly protect RNA from ISG20 degradation, but only if their thermodynamic stability is equal or superior to 20 Gibbs free energy (ΔG).

Overall, this study identifies two key features that mediate RNA protection from ISG20, shedding novel light on the complexity of the relationship between ISG20 and its RNA targets.

# Results

## ISG20 does not drive major changes in the cellular transcriptome, but leads to a small decrease in histone mRNAs

To determine the effects of ISG20 on cellular mRNAs at the whole-cell scale, the DNA coding ISG20 proteins was ectopically trans-fected in HEK293T cells, before WB and RNA-seq analyses 24 h later (Fig 1A and B for WB). The effects of WT ISG20 on the expression levels of cellular RNAs were compared with those of an ISG20 mutant presenting a single point mutation (M1 or D94A) within the conserved DEDD residue quartet that abrogates the ability of ISG20 to degrade RNA (Fig 1B) (17).

RNA-seq analysis indicated a clear separation of the samples by principal component analysis (Fig 1C); however, statistically sig-nificant changes in mRNA levels were contained within twofold and this independently from the levels of the expression of a given cellular RNA (Fig 1D and E), indicating that under the experimental conditions used here, ISG20 does not lead to major changes in the cellular transcriptome. However, when the threshold was lowered to a 1.5-fold change, respectively 208 and 230 genes appeared as up- and down-regulated in the presence of WT ISG20 (Fig 1F). These changes were not observed in the presence of the catalytically inactive M1 ISG20 mutant, indicating that transcriptional changes were linked to the expression of an active ISG20 (Fig S1A and Table S1). Interrogation of the Reactome database (28) with the genes down-regulated, and thus potentially targeted, by ISG20 indicated enrichment in histone-related pathways such as RMTs methylate histone arginines, HDACs deacetylate histones, RNA Polymerase I Promoter Opening (Fig S1B). Enrichment in these pathways was essentially driven by changes in the levels of 12 distinct histone-coding mRNAs (Fig 1F). Although cellular mRNAs are polyadenylated and through this feature associate with PABP1, all histone-coding mRNAs are non-polyadenylated and present instead a common stem–loop structure at their 3' end (Fig 1G and Table S2) (26). With the caveat that this RNA-seq analysis alone does not prove that ISG20 directly degrades histone mRNAs, these results opened up to two distinct but complementary questions: first, how most of the poly(A) cellular mRNAs could be protected by the action of ISG20, and second, why histone mRNAs that present a 3' terminal stem–loop that should protect them from ISG20 appear susceptible to this RNA exonuclease.

## ISG20 acts as a distributive enzyme

The behavior of viral or cellular RNAs depends on a plethora of factors that can overall influence the susceptibility to ISG20 and also potentially explain the results mentioned above. It appeared then of importance to turn to reductionist methods to identify and study the reasons for the behavior of cellular and viral RNAs toward ISG20.

To explore this question, we purified WT and M1 ISG20 (harboring a hexahistidine N-terminal tag and expressed/purified from a prokaryotic expression system, Fig S2) and characterized their behavior in biochemical assays. The first question we decided to address was whether ISG20 used a processive or a distributive

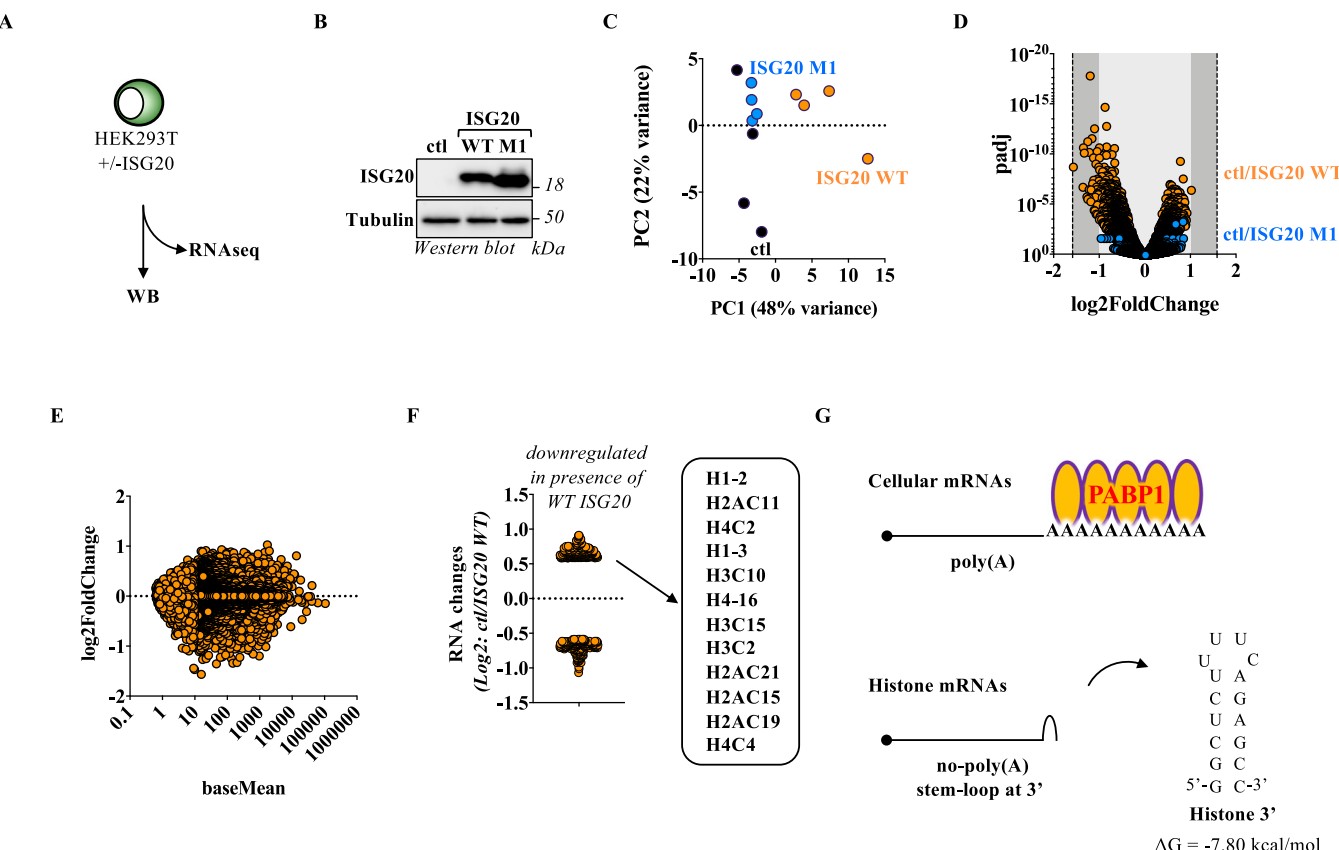

**Figure 1. ISG20 does not lead to overt changes in the cellular mRNA transcriptome with the exception of a small effect on histone mRNAs.**
**(A, B, C, D, E, F)** HEK293T cells were ectopically transfected with DNAs coding for control, WT, or M1 (D94A) ISG20 proteins, and 24 h later, cells were lysed and samples were analyzed by WB, using antibodies specific for the FLAG-tag present on ISG20 proteins or tubulin (B), or after ribosomal RNA depletion by RNA-seq (C, D, E, F). **(C)** Principal component analysis of the three conditions examined here. **(D)** mRNA changes in the control versus WT or M1 ISG20 conditions (*abscissa*) plotted in the function of the adjusted *P*-value (Padj, *ordinate*). **(E)** Analysis of the changes in individual genes (*ordinate*) as a function to the gene length (baseMean, *abscissa*). **(F)** Highlight of genes significantly modulated by ±1.5-fold in the presence of WT ISG20. **(G)** Schematic representation of histone mRNAs and stem–loop stability prediction using the UNAFold web server ([27]). The WB panels depict typical results obtained, whereas the graphs presented in the remaining panels present data obtained from four independent samples.
Source data are available for this figure.

---

mechanism for its exonuclease activity. This is a fundamental issue to apprehend the mechanism of action of ISG20, but this enzymatic property has never been properly analyzed. As simplified in the scheme of Fig 2A, the key difference between processive and distributive modes of RNA degradation relies on the fact that in the latter, the enzyme undergoes cycles of attachment and detachment from its target. As such, in this case once the radioactive-labeled RNA:protein complex is formed, RNA degradation can be decreased by the presence of an excess of unlabeled competitor RNAs, but instead this is not the case if the enzyme uses a processive mode of action. To this end, we conducted a steady-state processivity assay (Fig 2A) by incubating purified ISG20 with radiolabeled linear ssRNA (called "hot," 40 nucleotides long) in the presence, or absence, of an excess of cold ssRNA competitor in a reaction buffer containing $Mn^{2+}$ as detailed before. The addition of ISG20 triggered the exonucleolytic reaction, and the products of the reaction were monitored over 30 min by acryl–urea–PAGE analysis (Fig 2B). As expected, ISG20 degraded completely its RNA substrate over the 30 min of the assay. This activity was entirely dependent on the

presence of a functional catalytic site, given that the ISG20 M1 mutant did not degrade RNA substrates even after prolonged incubation times (Fig S3A). However, under the experimental conditions used here, WT ISG20-mediated degradation was completely prevented in the presence of the cold RNA competitor, indicating that ISG20 degrades RNA according to a distributive mode of action.

## ISG20-mediated degradation of a poly(A)-containing RNA target is prevented by PABP1

Given that the expression of ISG20 in cells does not result in overt cellular RNA degradation and that ISG20 is a distributive enzyme, we hypothesized that a *trans*-acting factor such as PABP1 could shield poly(A) cellular mRNAs from ISG20-mediated degradation.

First, we expressed and purified recombinant PABP1 (Fig S2), then verified its binding on a 5′ radiolabeled linear RNA containing 20 adenosines at its 3′ end by an electrophoretic mobility shift assay (EMSA; Fig 3A and B). As expected, a PABP1:RNA complex was formed in a manner proportional to the concentration of PABP1. Next, we

**A**

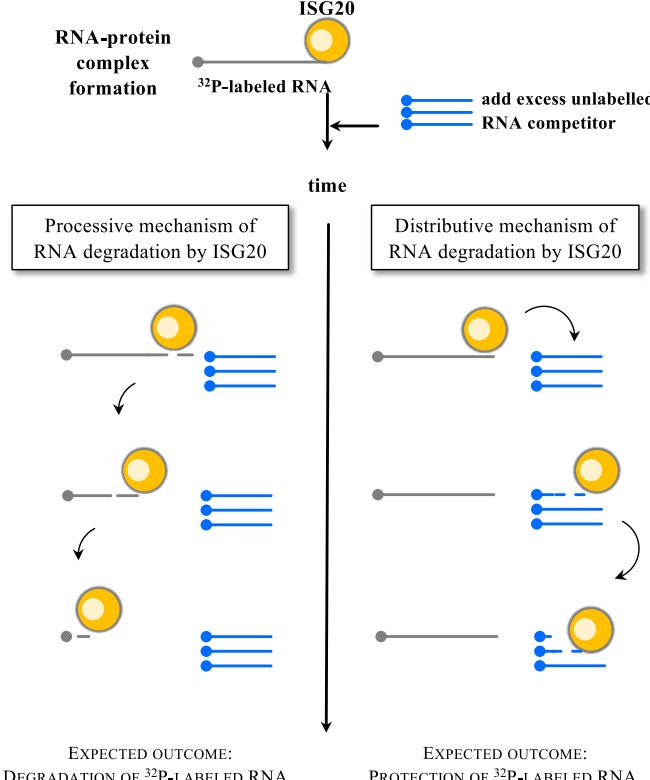

**B**

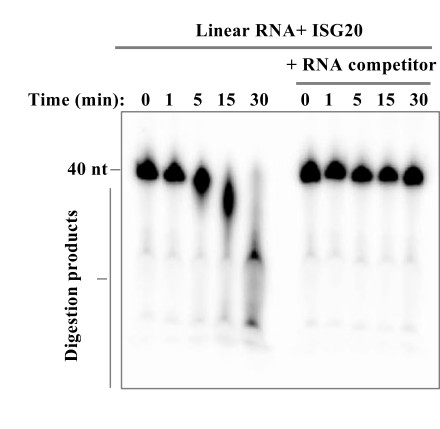

**Figure 2. ISG20 behaves as a distributive enzyme in vitro.**
**(A)** Schematic representation of the assay used to determine whether ISG20 acts as a processive or distributive enzyme. **(B)** Linear target RNA was incubated with ISG20 in the presence or absence of cold ssRNA, and the reaction was arrested at the indicated time before acryl–urea–PAGE analysis of ISG20-mediated digestion. The panel presents typical results out of three independent experiments.
Source data are available for this figure.

evaluated the possibility that PABP1 could interfere with ISG20-mediated degradation by shielding the poly(A) tail by performing a steady-state exonuclease assay in the presence of increasing concentrations of PABP1. The substrate was first incubated for 20 min with increasing concentrations of PABP1, then ISG20 was added, and its activity was monitored over a 1-h period. Under these conditions, a strong protection was exerted by PABP1 against the ISG20-mediated RNA degradation (Fig 3C and D). In contrast, ISG20 remained able to degrade its RNA target in control conditions (Fig 3C), in the presence of BSA, or when PABP1 was used on an RNA devoid of the poly(A) tail (Fig S3B and C). Overall, these data indicate that PABP1 is able to shield the 3' extremities of RNAs from ISG20-mediated degradation. Given that poly(A) tails are mandatorily present on cellular mRNAs, this likely represents the key feature of protection of cellular RNA from the exonuclease activity of ISG20.

### PABP1-bound cellular mRNA is resistant to ISG20-mediated degradation

To assess the protective role of PABP1 against cellular mRNA degradation by ISG20, we chose to avoid PABP1 silencing–based strategies in cells, as the pleiotropic roles of this protein in cellular

mRNA metabolism would have complexified the analysis of the results. Instead, we chose to purify cellular mRNA according to two distinct procedures (Fig 4A): in the first, cellular mRNAs were extracted from HEK293T cells by TRIzol and then directly purified on columns in the absence of cellular proteins; in the second, mRNAs were purified as a complex with PABP1 after immunoprecipitation with an anti-PABP1 antibody (Fig S3D). Under these conditions, when the two pools of mRNAs were incubated with ISG20 in vitro, only protein-free poly(A) mRNAs were susceptible to ISG20 degradation, whereas on the contrary, PABP1-bound mRNA resisted ISG20 (representative agarose gel panels and their densitometric quantification) (Fig 4B). Therefore, these results indicate that PABP1 is indeed able to protect cellular mRNAs from ISG20-mediated degradation.

### The Mopeia arenavirus is resistant to ISG20-mediated inhibition

To re-examine the possibility that stem–loop structures could indeed protect from ISG20 from a distinct angle, we studied the behavior of the Mopeia virus (MOPV) in A549 lung epithelial cells expressing a dox.-inducible form of ISG20. MOPV belongs to the *Arenaviridae* family and is a bisegmented ambisense RNA virus. The segmented portions of the genome of *Arenaviridae* code for genes

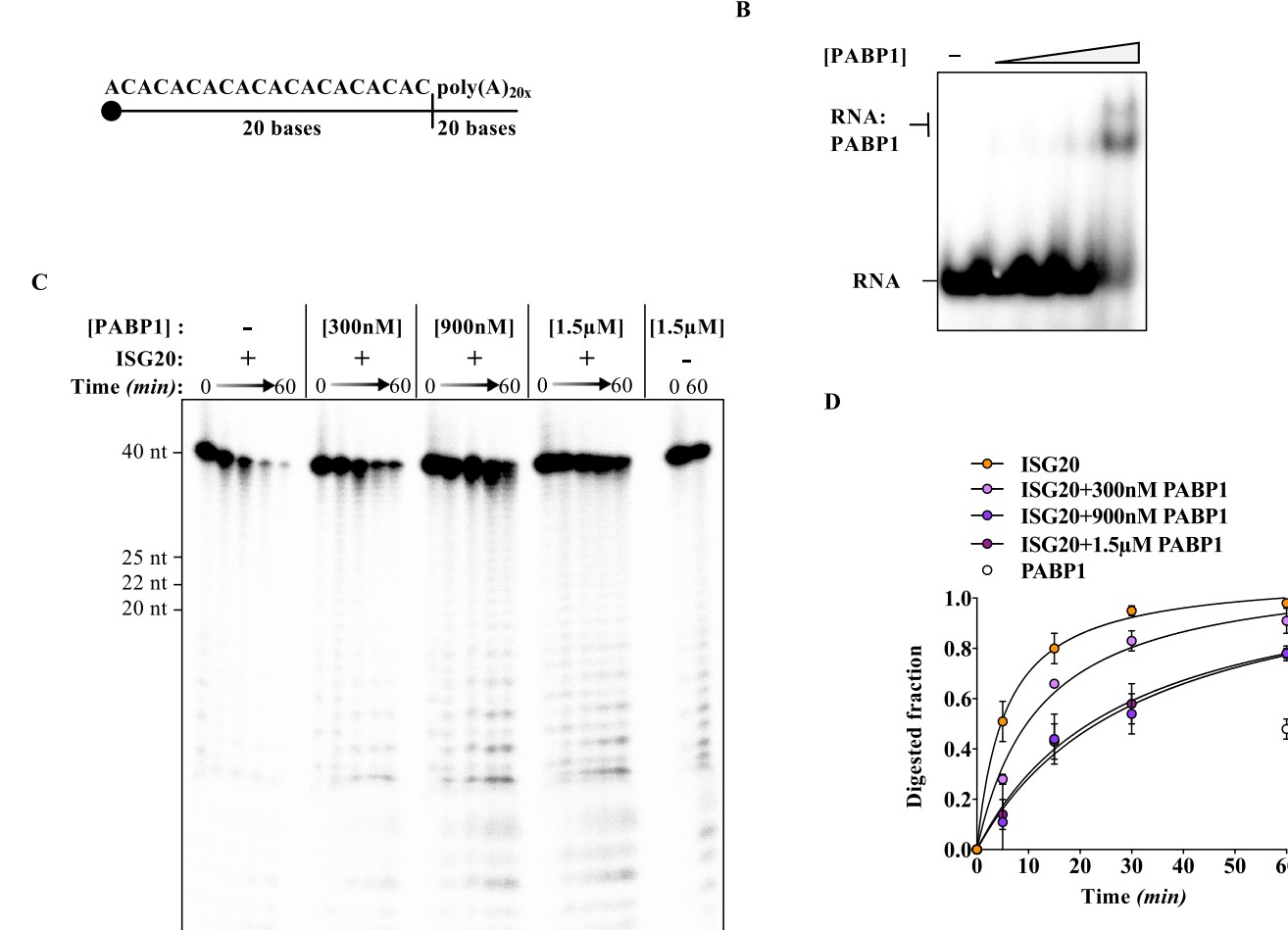

**Figure 3.  Poly(A)-binding protein (PABP1) protects poly(A)-tailed RNA from ISG20-mediated degradation.**
**(A)** ssRNA sequence used in the protein protection assay bearing a poly(A) 3′ tail of 20 nucleotides. **(B)** Representative electrophoretic mobility shift assay illustrating the interaction of PABP1 with $^{32}$P 5′ end-labeled ssRNA. The RNA substrate (10 nM) was incubated with increasing concentrations of PABP1 (50, 100, 200, and 500 nM) or without protein, under the conditions described in the Materials and Methods section. **(C)** Acryl–urea–PAGE analysis of the dose-dependent protection of PABP1 from the exonuclease activity of ISG20 on the indicated $^{32}$P-labeled ssRNA. **(D)** Quantification of the protein protection assay presented in (C). Graph showing the fractions of RNA digested by ISG20 as a function of time. The corresponding values for the digested fraction at time t (Dft) are calculated by the equation Dft = 1-(Rft/Rf0), where Rft is the intensity of the band corresponding to undigested RNA at time t and R0 at time 0, before the addition of the enzyme. Data are derived from three independent experiments (mean ± SD). The best-fitting equation was found with KaleidaGraph (Synergy Software) program. *$P < 0.05$ according to t tests (unpaired, two-tailed) between the conditions ISG20 alone versus ISG20 + PABP1 at the indicated time points.
Source data are available for this figure.

in opposite polarity that converge into a common genome portion named intergenic region (IGR). This region is characterized by long palindromic stretches that fold into highly stable stem–loop structures (Table S2) that are thus present at the 3′ of all viral RNAs (Fig 5A; the plus-strand orientation of the IGR sequences of the long and short segments of the MOPV genome, L and S respectively, is presented) (29, 30).

A549 cells expressing WT or M1 (D94A) ISG20 proteins were challenged with an MOI of 0.01 of MOPV, and both cells and supernatants were analyzed 48 h later to measure the accumulation of the viral protein Z by WB, or the accumulation of viral RNA in the cell and in virion particles released in the cell supernatant (Fig 5B and C). Under these conditions, ISG20 did not affect either the intracellular or the extracellular viral RNA quantities nor the levels of accumulation of the Z protein, indicating that MOPV is indeed resistant to ISG20 and

strongly supporting the notion that structured 3′ ends can indeed act as an element of RNA resistance of viruses to ISG20 (Fig 5C).

### 3′ stem–loops protect RNAs from ISG20-mediated degradation, depending on their stability

As the agnostic analysis of cellular RNA stability indicated that histone RNAs with a 3′ stem–loop structure are more sensitive to ISG20-mediated degradation than polyadenylated RNAs, we next decided to assess the behavior of stem–loops that decorate histone mRNAs and the IGR L of MOPV toward ISG20 in vitro. To this end, the indicated RNA substrates were radiolabeled, folded, and used as substrates for an ISG20-mediated steady-state exonucleolytic reaction (Fig 6A and B and Tables S2 and S3). Under these experimental conditions, the histone 3′ stem–loop exhibited an

**A**

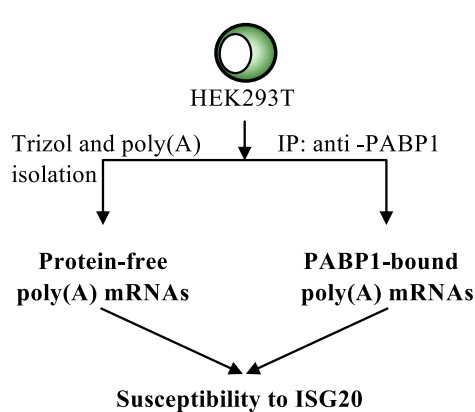

**B**

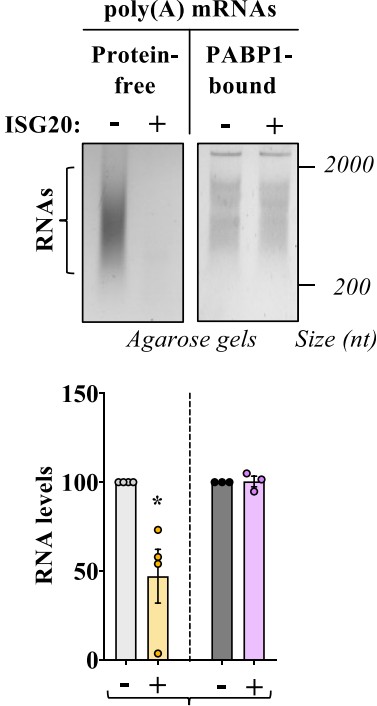

**Figure 4.  Poly(A)-binding protein (PABP1)–bound cellular mRNAs are protected against ISG20-mediated degradation.**
**(A)** Schematic representation of the assay used to determine the protective effect of PABP1 on cellular mRNAs. HEK293T cells were lysed and mRNAs extracted either after TRIzol and poly(A) bead purification or else after immunoprecipitation with an anti-PABP1 antibody. **(B)** RNAs obtained according to these two procedures were subjected to ISG20-mediated degradation for 45 min before migration on an agarose gel. The panels present representative agarose gels, and the graph presents densitometric quantification of the smear representing mRNAs ranging from 2,000 to 500–600 bp (n between 3 and 4; *$P < 0.05$, followed by one-way ANOVA with Dunnett's multiple comparison test).
Source data are available for this figure.

intermediate phenotype of partial resistance/susceptibility to ISG20 when compared to a linear RNA control, in line with the small but detectable effect of ISG20 on histone mRNAs measured in cells. On the contrary, and in line with the full resistance of MOPV replication in ISG20-expressing cells, the MOPV-derived stem–loop was completely resistant to ISG20 (Fig 6B). In this case, the RNA stem–loop structure gave rise to three distinct forms upon migration in an acrylamide gel that we have not further investigated and that we believe to correspond to different RNA conformers (31). The results obtained on the apparent higher susceptibility of histone mRNAs to ISG20 and the converse resistance of MOPV appear contradictory. However, the inspection of the thermodynamic stability of the stem–loop present on histone versus MOPV RNAs reveals important differences that could themselves be at the basis of their different behaviors.

To explore this more formally and at a more general scale, we compared three well-characterized stem–loop structures derived from the transactivating region of the HIV-1 genome (TAR) and specifically the full TAR structure (TAR, spanning nucleotides 454–512 of the HIV-1 provirus [32]), along with two modified versions, TAR-9SL and TAR-5SL, in which the size of the stem has been reduced to nine and five base pairs (Fig 7A and Table S4; adapted from UNAFold [27]). These structures were chosen because the folding

thermodynamics computationally calculated at −32.90, −23.00, and −9.80, kcal/mol, respectively, cover the spectrum of stem–loop stability that can be measured in histone, as well as MOPV mRNAs, thereby allowing us to define the boundaries of thermodynamic stability with which stem–loops can provide a sufficient protection against ISG20-mediated degradation.

The different RNA substrates were radiolabeled, folded, and purified, before use in an ISG20-mediated steady-state exonucleolytic reaction, as previously described in the Materials and Methods section. We observed a lower intensity of labeling of the TAR structure compared with others, which was not investigated further, but that we believe because of the lower efficiency of T4 polynucleotide kinase activity on this substrate. As expected, the linear RNA substrate was degraded to completion very rapidly (Fig 7B). In contrast, the most stably structured RNAs (TAR-9SL and TAR) were completely resistant to ISG20 cleavage (Fig 7B). Instead, the TAR-5SL RNA presented a more complex pattern of digestion. In particular, a first ladder of five nucleotides corresponding to the degradation of the basal stem of the RNA appeared after 5 min of reaction, followed by the subsequent digestion of the 4-bp-long apical stem and the complete digestion of target RNA at the end of the reaction. As such, these results indicate that stem–loop

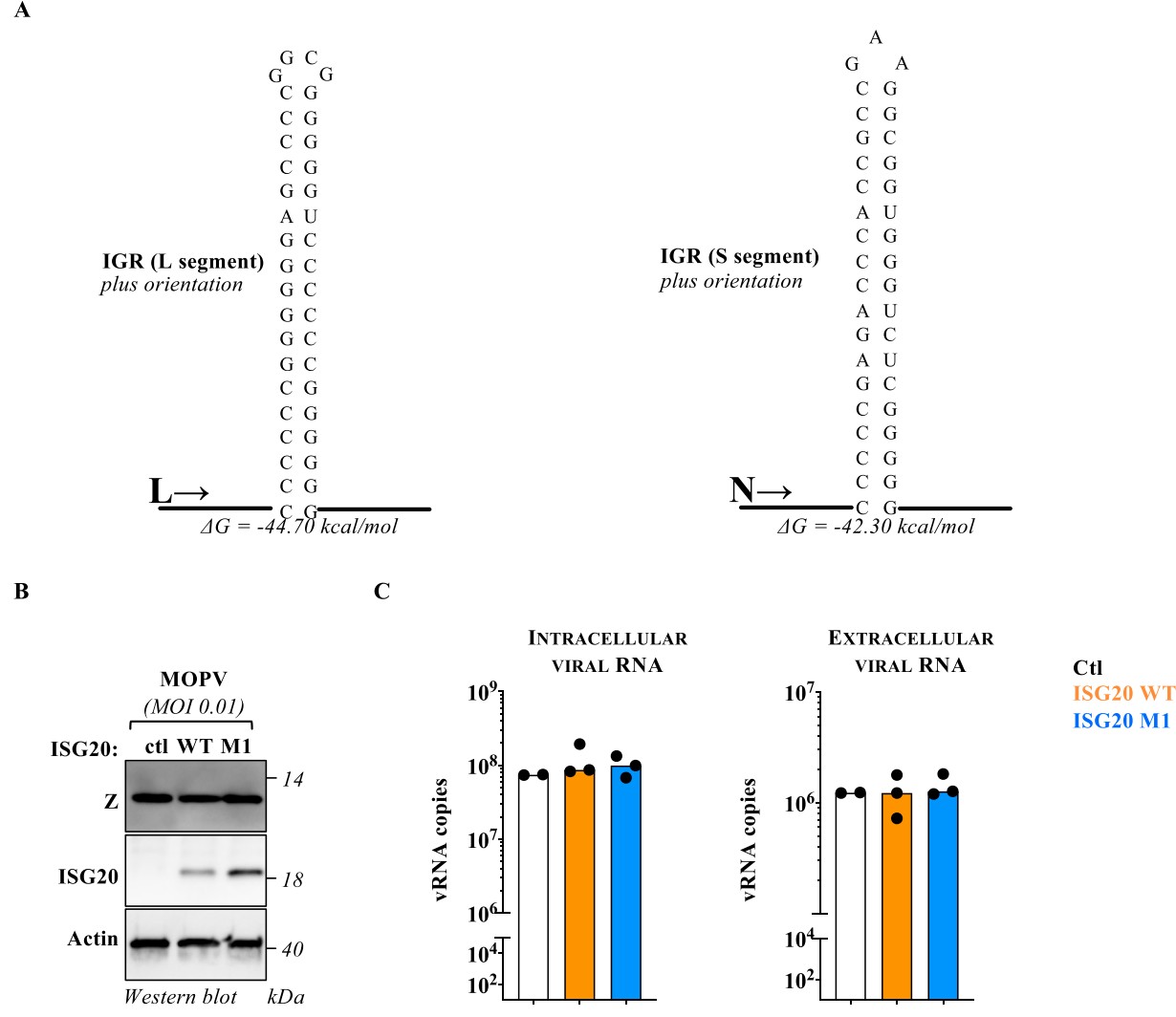

**Figure 5. Mopeia arenavirus is resistant to ISG20-mediated inhibition.**
**(A)** RNA structures of the intergenic regions of the L and S segment of the Mopeia virus, as determined by RNAFold. **(B, C)** A549 cells stably expressing WT or M1 ISG20 proteins were challenged with an MOI of 0.01 of Mopeia virus, and both cells and supernatants were analyzed 48 h later by WB (using antibodies specific for the viral protein Z, the FLAG-tag present on ISG20 proteins, or actin) and RT–qPCR. The WB panel presents typical results obtained, whereas the graphs present data obtained from three independent experiments carried out in duplicate (two for control). Differences between control and remaining samples were not statistically significant, followed by one-way ANOVA with Dunnett's multiple comparison test.
Source data are available for this figure.

structures must possess a minimal stability to provide full protection against ISG20, whereas short stem–loop remains susceptible to it. Of note, the lack of digestion of stable stem–loop structures is unlikely to be due to the absence of binding to ISG20, because ISG20 was able to associate with TAR-9SL in a classical EMSA (Fig S4), suggesting that ISG20 bears the capability to sample RNAs even if they are highly structured.

## Discussion

In this study, we combined whole-cell RNA-seq approaches with biochemical characterization of the behavior of model RNAs toward ISG20 and we identified two novel elements of protection of RNA against ISG20-mediated degradation: poly(A) tails complexed with PABP1 and stem–loop RNA structures of a minimal stability that we define here.

Contrarily to the 2′–5′ oligoadenylate-dependent ribonuclease L (RNaseL), which is produced in an inactive form and is activated only in the presence of 5′-phosphorylated, 2′,5′-linked oligoadenylates, themselves produced by the 2′–5′ oligoadenylate synthase (OAS) (reviewed in reference 33), ISG20 does not seem to be produced in an inactive form, because ISG20 directly purified from cellular lysates behaves as a highly efficient RNase (17). Although it remains possible that protein cofactors lost during the purification procedure may control ISG20 in the cell, whether such cofactors exist remains unclear. Similarly, ISG20 targeting sites of viral

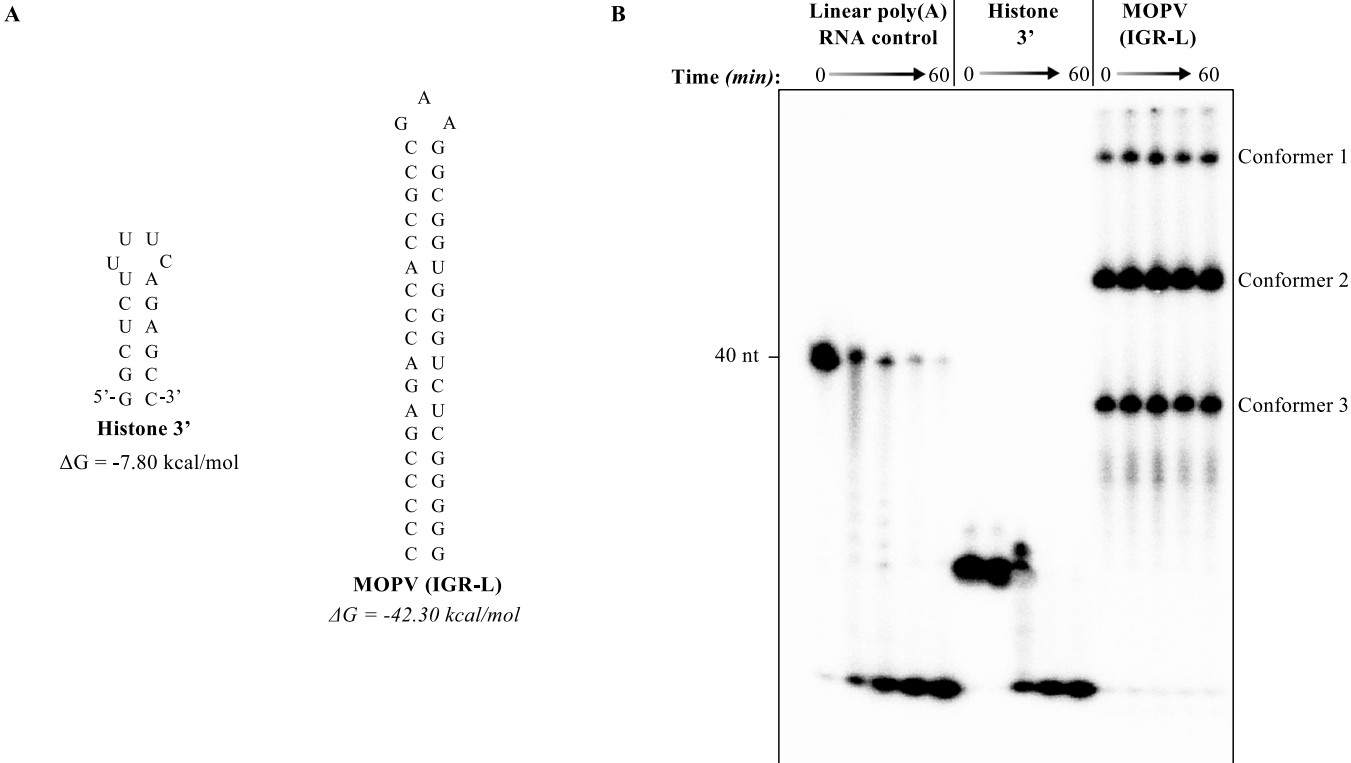

**Figure 6. Analysis of the behavior of histone- and Mopeia virus–derived 3′ RNA stem–loop structures.**
**(A)** Structural folding and thermodynamic stabilities of the indicated RNAs predicted with UNAFold web server. **(B)** Time course of ISG20-mediated degradation of the indicated [32]P-labeled RNAs, before analysis by acryl–urea–PAGE. The gel depicts typical results obtained from two independent experiments.
Source data are available for this figure.

replication, such as viral factories, could be envisioned as a mechanism to specifically restrict its RNase activity against viral RNAs. However, this possibility has not been documented in the literature and it is unlikely to explain how ISG20 can target viruses with very diverse replication modes. As such, how the cell restrains and directs ISG20 specifically against viruses remains unclear.

An alternative mechanism of viral inhibition that has been proposed for ISG20 indicates that ISG20 does not degrade viral RNA directly, but rather inhibits its translation. Given that in most cases, viral RNAs end up being massively translated in the cell, this mechanism would not need to specifically target viral RNAs, but simply the most translated RNA species. The mechanism through which this occurs is unclear. However, given that even according to this mechanism of viral inhibition, the RNase activity of ISG20 remains of key importance, this suggests that ISG20 must target an RNA, cellular in this case, raising again the question of how ISG20 is able to discriminate one RNA from the other.

Recent studies have suggested that epitranscriptomic modifications may either protect 2′-O nucleotide modifications on the HIV-1 genome (22) or expose (m6A on hepatitis B virus) RNAs from ISG20-mediated degradation (19), raising the general question of how viruses and cells can exploit epitranscriptomic modifications to influence the activity of ISG20. The role that such modifications may bear for the susceptibility of target RNAs to ISG20 has not been explored directly in this study. Yet, it remains for the moment

unclear how they could provide a discriminating signal for ISG20, given their abundance on not only cellular but also viral RNAs, as an increasing number of studies are highlighting (34). Besides, it remains unclear how epitranscriptomic modifications that often occur at internal positions of the RNA can act distally to protect the 3′ end of the RNA that is attacked by ISG20.

Our RNA-seq analyses indicated that ISG20 induces only minor changes in the cell transcriptome under the experimental conditions used here, with the most apparent being a decrease in histone mRNAs. We believe this is likely to lead to chromatin changes that in turn drive the minor transcriptomic variations observed here, although we do not know whether these changes would also be observed in the context of IFN responses in which ISG20 is normally expressed. However, these changes are not observed in the presence of a catalytically inactive mutant, indicating that they are linked to the RNase activity of ISG20, and we have therefore used these results as a starting point to identify novel RNA elements that could modulate the susceptibility to ISG20.

Given that histone mRNAs are the only cellular mRNAs devoid of poly(A) tail and possess instead a short stem–loop at their 3′ end, we investigated the weight of these two features in the susceptibility of target RNAs to ISG20 in vitro. Our results indicate that the poly(A) tail is not a protective element per se, but becomes so by acting as a docking site for PABP1, a protein that associates with the poly(A) tail present in cellular mRNAs (35). These results thus

**A**
**B**

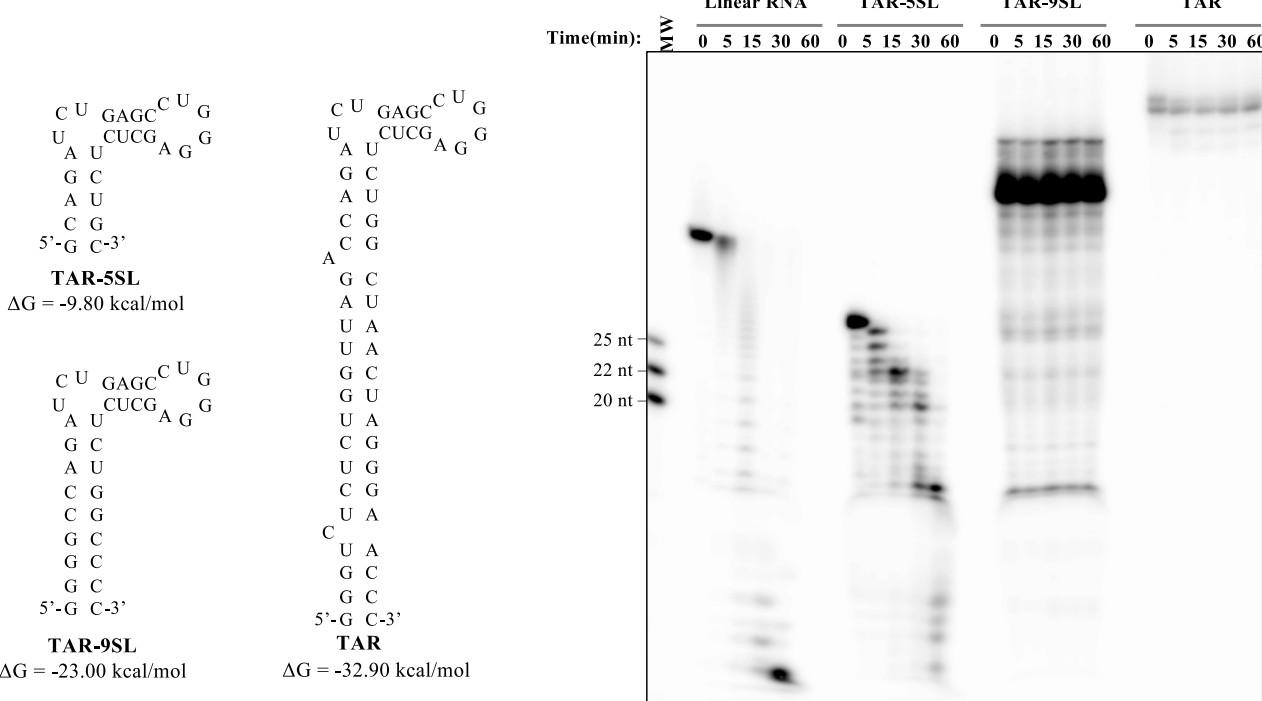

**Figure 7. Only high-energy structured 3′ RNAs provide protection from ISG20.**
**(A)** Structural model of dsRNAs used here that differ for the length of their stem portions predicted with UNAFold web server (27) and Table S3. **(B)** Time course of ISG20-mediated digestion of linear single-stranded RNA (ssRNA, i.e., $(AC)_{10}A_{20}$) and the indicated stem–loop structures. The exonuclease reaction was analyzed by acryl–urea–PAGE under the conditions described before. The gel depicts typical results obtained out of three independent experiments. Source data are available for this figure.

provide a simple explanation of why cellular mRNAs are essentially spared from ISG20.

The second element that our study identified, in line with previous results in the literature (4, 14), is constituted by stem–loop structures and more precisely double-stranded regions that can protect RNAs by sequestering free 3′ ends from ISG20, even in the absence of a poly(A) tail and its associated proteins. In line with the RNA-seq analysis, the stem–loop present at the 3′ of non-polyadenylated histone mRNA provides a partial protection from ISG20. Building on this result, we have evaluated more broadly the relationship existing between terminal stem–loop stability and susceptibility to ISG20 and we determined that thermodynamic stabilities comprised between −9.8 and −23 kcal/mol constitute the boundary between susceptibility and resistance to ISG20. These values ought to be considered as a first simplified reading of the action of ISG20, because these structures can be themselves regulated by the dynamic association with proteins in cells, as is the case of the stem–loop-binding protein (26) that binds to the 3′ end of histone mRNAs.

When considering the spectrum of viruses that has been described to be susceptible to ISG20 (7, 8, 9, 10, 11, 12, 13, 14, 15, 16, 17, 18, 19, 20, 21), epitranscriptomic modifications, coated poly(A) tails, and/or stem–loop structures cannot provide a unique explanation through which ISG20 may distinguish viral from cellular RNAs. Indeed, certain viruses also use epitranscriptomic modifications to regulate the metabolism of their RNAs; a large number of viruses

code for proteins that coat the viral genome and that may also help shielding it from ISG20, similar to PABP1 on poly(A) tails; and certain viruses can protect their RNA by sequestering it in specific cellular compartments. As such, the identification of elements that can modulate the susceptibility or resistance of given RNAs toward ISG20 remains of key importance.

To summarize, our study uncovers poly(A) tails coated with PABP1 and stem–loop structures with a minimal thermodynamic stability as two novel elements that can protect target RNAs from the RNase activity of ISG20. These results thus contribute to shed new light on the complexity that underlines the balance between susceptibility and resistance of RNA molecules to ISG20.

# Materials and Methods

### Cell culture and DNA constructs

Human kidney epithelial HEK293T and lung epithelial A549 cells (CRL-3216 and CCL-185; ATCC, respectively) were maintained in complete DMEM (cat. 61965-026; Gibco) supplemented with 10% FBS (cat. F7524; Sigma-Aldrich) and 100 U/ml of penicillin–streptomycin (cat. 11548876; Gibco). Transient transfections were performed using an in-house calcium phosphate (calcium HBS) method (36). Two

series of plasmids coding for an N-terminal FLAG-tagged ISG20 (*WT* and D94A mutant, referred to as WT and M1, respectively) were used, as previously described (5, 17, 22): pcDNA3.1+-based plasmids used for ectopic expression and pRetroX-tight-Puro–based ones (cat. PT3960-5; Clontech) used here to generate stable cell lines expressing ISG20 upon induction with doxycycline (dox.), after retroviral-mediated gene transduction (see below).

The following antibodies were used for Western blotting: anti-FLAG (cat. F3165; Sigma-Aldrich), anti-α-tubulin (cat. T5168; Sigma-Aldrich), anti-GFP (cat. G1544; Sigma-Aldrich), and anti-mouse and anti-rabbit IgG peroxidase-conjugated (cat. A9044 and cat. AP188P; Sigma-Aldrich, respectively).

## Generation of stable cell lines

Murine leukemia virus (MLV) retroviral particles were obtained by transient transfection of HEK293T cells with plasmid DNAs coding for the following: the MLV structural protein Gag-Pro-Pol (pTG5349) (37); the envelope protein G from the vesicular stomatitis virus (VSVg); and the mini-viral genome coded by the pRetroX-Tight-Puro (8:4:8 µg DNA per 10-cm plate dish, respectively). Parallel virus production was obtained by substitution of the ISG20-coding viral genome with pRetroX-Tet-On (cat. 632104; Clontech) that codes for the TetOn protein and allows for the dox.-inducible expression of ISG20. Transfection with these three plasmids allows the production of MLV-derived retroviral particles that can be used to obtain stable insertion of the transgene of interest into the genome of target cells. Viral particles released in the supernatant of transfected HEK293T cells were harvested 24–48 h after transfection, syringe-filtered (0.45 µm), and used to challenge A549 cells. After transduction with both types of particles (pRetroX-Tight-Puro-ISG20 and pRetroX-Tet-On), cells were selected as a pool upon incubation with puromycin and G418 (cat. P9620 and G418-RO; Sigma-Aldrich). ISG20 expression was routinely induced upon incubation of cells for 18 h with 1.5 µg/ml of doxycycline (cat. 631311; Takara Bio).

## Mopeia virus infections

A549-ctl, A549-WT, or A549-M1 cells were plated at 250,000 cells per well in 24-well plates in the presence of doxycycline (1.5 µg/ml) 24 h before infection with a recombinant MOPV-WT (strain AN21366) (38) at an MOI of 0.01. Supernatants and cells were collected 48 h later for further analyses. Viral loads in cells and cell supernatants were measured by RT–qPCR. To this end, RNAs were extracted using a QIAamp Viral RNA kit (cat. 52904; Qiagen) and amplified using the SensiFAST Probe No-ROX One-Step kit (cat. BIO-76001; Bioline) using primers and probe targeting the NP gene of MOPV (Table S5). Cellular RNAs were instead extracted using an RNeasy mini extraction kit (cat. 74004; Qiagen) and then treated as described above. The antibodies used in this study were as follows: rabbit anti-Z (produced in-house), HRP-conjugated anti-FLAG (cat. A8592; Sigma-Aldrich), and anti-actin (cat. A3854; Sigma-Aldrich).

## RNA-seq

HEK293T cells were seeded in 10-cm dishes, and cells were transfected 24 h later with 10 µg of a pcDNA3.1+ control vector or of a vector expressing FLAG-ISG20 WT or M1, along with a vector coding for a GFP reporter that we used as an internal control. 24 h after transfection, total RNAs were extracted using TRIzol added directly on the culture plate. Extracted RNAs were depleted from ribosomal RNAs using antisense DNA oligonucleotides complementary to rRNA and RNaseH as previously described (39), followed by DNase treatment to remove DNA oligonucleotides. High-throughput sequencing libraries were prepared as described (40). Briefly, RNA samples depleted from ribosomal RNAs were fragmented using RNA fragmentation reagent (cat. AM8740; Ambion) for 3 min and 30 s at 70°C followed by inactivation with the provided "stop" buffer. Fragmented RNAs were then dephosphorylated at their 3′ end using T4 polynucleotide kinase (PNK, cat. M0201; New England Biolabs) in MES buffer (100 mM MES–NaOH, pH 5.5, 10 mM $MgCl_2$, 10 mM β-mercaptoethanol, and 300 mM NaCl) at 37°C for 3 h. RNA fragments with a 3′-OH were ligated to a preadenylated DNA adapter. After this, ligated RNAs were reverse-transcribed with Superscript III (cat. 18080044; Invitrogen) with a barcoded reverse transcription primer that anneals to the preadenylated adapter. After reverse transcription, cDNAs were resolved in a denaturing gel (10% acrylamide and 8 M urea) for 1 h and 45 min at 35 W. Gel-purified cDNAs were then circularized with CircLigase I (cat. CL4111K; Lucigen) and PCR-amplified with Illumina's paired-end primers 1.0 and 2.0. PCR amplicons (12–14 cycles for RNA-seq and 4–6 cycles for ribosome profiling) were gel-purified and submitted for sequencing on the Illumina HiSeq 4000 platform.

## Analysis of high-throughput sequencing reads

Sequencing reads were split with respect to their 5′ in-line barcode sequence. After this, 5′-barcode, 6 nt Unique Molecular Identifiers were removed from reads using an in-house script. After this, the first seven nucleotides at the 5′ end of R1 reads and the first 60 nt at the 3′ end of R2 reads were further removed to avoid dovetails. 3′ adapter sequences were removed using Trimmomatic (41) with the following parameters "PE MAXINFO:36:0.2." To remove rRNA and tRNA contaminant sequences, reads were first mapped to a custom set of sequences including human 45S and 5S rRNA, tRNAs, phiX174, and GFP sequences using Bowtie2/2.3.3 (42) with the following parameters "bowtie2 -t --sensitive."

Reads that failed to map to this custom set of sequences were next aligned to the human hg38 assembly and the GRC-Genecode GRCh38.v37 primary assembly annotation using HISAT2 2 (v2.1.0) (43) with the following parameters "hisat2 -t --no-unal --phred33 -p 16 -k 10 --min-intronlen 20 --max-intronlen 1000000 --rna-strandness RF I 100 -X 700 --fr." Read counts on all transcripts of interest were obtained using the HTSeq-count package (44) with the following parameters "htseq-count -f sam -r pos -s yes -a 10 --nonunique=none -m union."

## Differential analysis with DESeq2

Differentially expressed genes upon ISG20 overexpression were obtained using DESeq2 (45) (version 1.24.0) in R (version 3.6.3).

## Protein expression and purification

The *WT* ISG20 full coding sequence (amino acids 1–181) was cloned into the pPROEX HTa vector (cat. 10711018; Thermo Fisher Scientific)

by standard molecular biology techniques. Protein expression from this plasmid results in the presence of an N-terminal hexahistidine tag–ISG20 fusion. The pTRC-HisA plasmid harboring the PABP1-coding sequence was kindly provided by Dr. Théophile Ohlmann (CIRI, Lyon, France).

ISG20 was expressed in *E. coli* BL21 (DE3) Rosetta/pLysS strain (cat. EC0114; Thermo Fisher Scientific and cat. 71403; Sigma-Aldrich, respectively). Bacterial cells were grown in an LB medium for 16 h at 25°C, and then, protein expression was induced with 0.1 mM IPTG (isopropyl-$\beta$-D-thiogalactopyranoside, cat. PHG000110; Sigma-Aldrich). Cultures were incubated for 3 h 30 min at 25°C under continuous shaking, then harvested by centrifugation for 15 min at 5,000$g$. Each pellet was resuspended with buffer A (50 mM Tris–HCl, pH 7.5, 150 mM NaCl, 20 mM imidazole, 5 mM $\beta$-mercaptoethanol, and 20% glycerol, vol/vol) supplemented with 1 mM of ATP, 1 mM MgCl$_2$, and Protease Inhibitor Cocktail Tablet (cat. 11836170001; ROCHE, cOmplete Sigma-Aldrich). The soluble lysate was applied to a prepacked nickel column (HisTrap HP column, cat. GE17-5247-01; Cytiva Europe GmbH) and fractionated on an AKTA pure system (Cytiva Europe GmbH) using a linear gradient from buffer A to buffer B (buffer A supplemented with 0.5 M imidazole) over 10 column volumes. A second step of purification was carried out using a Superdex 75 10/300 GL column (Cytiva Europe GmbH) with an isocratic elution carried out with storing buffer (50 mM Tris–HCl, pH 7.5, 50 mM NaCl, 2 mM $\beta$-mercaptoethanol, and 20% glycerol, vol/vol). Finally, glycerol was added to reach the concentration of 50% and proteins were stored at −20°C.

Bacterial cells for His-tagged PABP1 expression were grown in the LB medium at 37°C before inducing protein expression overnight at 20°C with 1 mM IPTG and shifting the culture to 20°C. The pellet was resuspended in a lysis buffer composed of 50 mM Tris–HCl, pH 7.5, 300 mM NaCl, 20 mM imidazole, 5 mM $\beta$-mercaptoethanol, and 20% glycerol, and purification was performed as described above. A second step of purification was carried out using a Superdex 75 10/300 GL column with high ionic stringent buffer (50 mM Tris–HCl, pH 7.5, 1 M NaCl, 2 mM $\beta$-mercaptoethanol, 0.1 mM EDTA, and 20% glycerol). The protein was dialyzed against storing buffer (50 mM Tris–HCl, pH 7.5, 50 mM NaCl, 2 mM $\beta$-mercaptoethanol, and 20% glycerol, vol/vol), then stored at −80°C.

### In vitro RNA synthesis, purification, and radiolabeling

RNAs were produced after transcription from partially double-stranded DNA templates using the T7 RNA polymerase enzyme mix (cat. EP0112; Thermo Fisher Scientific), following the manufacturer's instructions and as described in reference 46. TAR (nucleotides 454–512 of the HIV-1 proviral genome # AF004394) and TAR-9SL DNA templates were produced by the annealing of two complementary oligonucleotides (Tables S5 and S6). RNAs were dephosphorylated using Calf Intestinal Alkaline Phosphatase (cat. M0525; New England Biolabs), then purified on denaturing 10% (wt/vol) polyacrylamide gel (29:1) as previously described (46). Before use in binding and enzymatic studies, dsRNA was heated in a refolding buffer (20 mM Hepes, pH 7.5, 0.2 M NaCl, 2 mM MgCl$_2$, and 2 mM DTT) for 3 min at 95°C followed by 40 min of slow controlled cooling to RT, and finally placed on ice. TAR-5SL and poly(A) RNAs

were chemically synthesized (Dharmacon). Radiolabeling of 50 pmol of the RNA substrate was performed in 20 $\mu$l of reaction mix using 10 U of T4 PNK (cat. M0201; New England Biolabs) and 3 $\mu$l of $\gamma^{32}$P-ATP (PerkinElmer) in 1X PNK buffer. The mixture was incubated at 37°C for 1 h before inactivating the reaction by incubating at 65°C for 20 min. Radiolabeled RNA was first purified using a MicroSpin G25 column (cat. GE27-5325-01; Sigma-Aldrich) following the manufacturer's instructions, then extracted with an equal volume of phenol:chloroform:isoamyl alcohol mix (25:24:1; cat. 516726-1SET; Sigma-Aldrich) precipitated with ethanol.

### EMSA

Samples were prepared by mixing a radiolabeled RNA (10 nM) with recombinant ISG20, as indicated in the text, or in the legend, in a buffer containing 50 mM Tris–HCl, pH 7.4, 150 mM NaCl, 5 mM $\beta$-mercaptoethanol, 10 mM MnCl$_2$, 0.1 $\mu$g/$\mu$l of BSA, and 5% (vol/vol) glycerol. The samples were incubated at 30°C for 30 min before being resolved by native 6% polyacrylamide (19:1) gel electrophoresis and analyzed by phosphorimaging. For PABP1 EMSA, 10 nM of poly(A)-containing substrate was incubated with PABP1 (50, 100, 200, and 500 nM) in a buffer containing 25 mM Tris–HCl, pH 7.4, 5 mM MgCl$_2$, 100 mM NaCl, 2.5 mM DTT, 0.2 $\mu$g/$\mu$l of BSA, and 10% (vol/vol) glycerol. Protein and RNA were incubated at 37°C for 15 min before being resolved by native gel, as previously indicated.

### 3′–5′ exonuclease assays

The nuclease assay was performed by mixing 5 nM of recombinant ISG20 with 500 nM of radiolabeled RNA substrate in a buffer containing 50 mM Tris–HCl, pH 7.4, 2.5 mM MnCl$_2$, 1 mM $\beta$-mercaptoethanol, 0.4 mM DTT, 0.1% (vol/vol) Triton X-100, and 10% (vol/vol) glycerol. The reaction was incubated at 37°C for 1 h. At the end of each incubation time, 3 $\mu$l aliquots were withdrawn and rapidly mixed with 3 $\mu$l of stop buffer containing 5 mM EDTA, 0.5% (wt/vol) SDS, 34% (wt/vol) glycerol, 0.5 M urea, 1% (wt/vol) formamide, 0.01% (wt/vol) xylene cyanol, and 0.01% (wt/vol) bromophenol blue. The collected samples were heated at 98°C for 5 min, then resolved in 15% polyacrylamide (ratio 19:1)–7 M urea gel, and analyzed by phosphorimaging using Typhoon FLA 9500 (Cytiva Europe GmbH). When indicated, the bands were quantified using ImageQuant software (Cytiva Europe GmbH) and the results analyzed by KaleidaGraph (Synergy Software).

Processivity tests were performed using cold RNA to trap the excess enzyme. ISG20 was previously incubated with a radiolabeled RNA substrate for 5 min, and then, 100 M excess of cold RNA substrate was added when indicated. Exonuclease reactions were subsequently performed as described above.

### PABP1 protection exonuclease assays

Thirty nanometers of RNA substrate was mixed with concentrations of recombinant PABP1 ranging from 50 to 500 nM and incubated for 20 min at 37°C in a buffer containing 10 mM Tris–HCl, pH 7.4, 10 mM MnCl$_2$, 1 mM DTT, 100 mM NaCl, and 10% glycerol. ISG20 was added at a final concentration of 120 nM, and the enzymatic assay was conducted as described above.

### Cellular mRNA isolation, immunoprecipitation, and ISG20 digestion

Total RNAs were extracted using TRIzol according to the manufacturer's instructions (cat. 15596018; Invitrogen). mRNAs were then isolated using PolyATtract mRNA Isolation Systems following the instruction (cat. Z5310; Promega). RNA was quantified by NanoDrop, and the number of mRNA molecules has been roughly estimated, assuming an average mRNA length of 1,200 nt. A nuclease assay was performed in 20 μl of reaction with 180 nM of mRNA and 7.2 μM of ISG20 (for a 4:1 molecule ratio between ISG20 and RNA) for 1 h at 37°C. Samples were then mixed with 2X loading buffer (95% formamide, 0.025% SDS, 0.025% bromophenol blue, 0.025% xylene cyanol, and 0.5 mM EDTA), heated at 70°C for 5 min, and then resolved on a 1% agarose gel. For the isolation of PABP1-bound mRNAs, HEK293T cells were lysed in RIP buffer (10 mM Tris–HCl, pH 7.5, 100 mM NaCl, and 0.5% NP-40) supplemented with 40 U/ml RNasin and Protease Inhibitor Cocktail (ab270055; Abcam) for 30 min on a nutator at 4°C. All processing steps were carried out at 4°C. After two centrifugations to clarify them, lysates were incubated for 1 h with 3 μg of rabbit anti-PABPC1 antibody per 10-cm dish plate (cat. 10970-1-AP; Proteintech), followed by incubation for another hour with G protein Dynabeads (cat. 10004D; Thermo Fisher Scientific). Beads were then washed three times at 4°C for 10 min with RIP buffer and then resuspended in a digestion buffer containing 10 mM Tris–HCl, pH 7.5, 10 mM $MnCl_2$, 1 mM DTT, 100 mM NaCl, and 10% glycerol. RNA was then treated as specified above.

### Statistical analyses

Statistical analyses were calculated with GraphPad Prism 8 or Excel software: t tests (unpaired, two-tailed), or one-way ANOVA with Dunnett's multiple comparison tests, as indicated in the legend of the relevant figures.

## Data Availability

All data are incorporated into the article and its online supplementary material. RNA-seq data are available in the Gene Expression Omnibus database of the NCBI as the GEO Submission number GSE233792.

## Supplementary Information

## Acknowledgements

We are grateful to Sylvain Baize for providing MOPV reagents. This work was supported by the Agence Nationale de Recherche (ANR, grant number ANR-20-CE15-0025-01 to A Cimarelli, EP Ricci, and S Deymier). F Fiorini is supported by the CNRS (French National Centre for Scientific Research) and by FINOVI Foundation (Contract number 247479). Funding for the open access charge was provided by Agence Nationale de Recherche.

## Author Contributions

C Louvat: investigation and methodology.
S Deymier: formal analysis, investigation, and methodology.
X-N Nguyen: investigation.
E Labaronne: formal analysis.
K Noy: investigation.
M Cariou: data curation.
A Corbin: data curation.
M Mateo: supervision, investigation, and methodology.
EP Ricci: conceptualization, supervision, funding acquisition, and investigation.
F Fiorini: conceptualization, supervision, funding acquisition, investigation, and writing—review and editing.
A Cimarelli: conceptualization, supervision, funding acquisition, project administration, and writing—original draft, review, and editing.

## Conflict of Interest Statement

The authors declare that they have no conflict of interest.

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
