## [Reviewer comments · Life Science Alliance]

Life Science Alliance

Stable Structures or PABP1 loading protect cellular and viral RNAs against ISG20-mediated decay

Camille Louvat, Severine Deymier, Xuan-Nhi Nguyen, Emmanuel Labaronne, Kodie Noy, Marie Cariou, Antoine Corbin, Mathieu Mateo, Emiliano Ricci, Francesca FIORINI, and Andrea Cimorelli

DOI: <https://doi.org/10.26508/lsa.202302233>

Corresponding author(s): *Andrea Cimorelli, International Center for Infectiology Research*

Review Timeline:

Submission Date:	2023-06-23
Editorial Decision:	2023-07-24
Revision Received:	2024-01-30
Editorial Decision:	2024-02-22
Revision Received:	2024-02-23
Accepted:	2024-02-23

Scientific Editor: *Eric Sawey, PhD*

Transaction Report:

July 24, 2023

Re: Life Science Alliance manuscript #LSA-2023-02233-T

Andrea Cimorelli
CNRS/ENS-Lyon/Université Lyon
Human Virology
46, allée d'Italie
Lyon 69007
France

Dear Dr. Cimorelli,

Thank you for submitting your manuscript entitled "Stable Structures or PABP1 loading protect cellular and viral RNAs against ISG20-mediated decay" to Life Science Alliance. The manuscript was assessed by expert reviewers, whose comments are appended to this letter. We invite you to submit a revised manuscript addressing the Reviewer comments.

Thank you for this interesting contribution to Life Science Alliance. We are looking forward to receiving your revised manuscript.

Sincerely,

B. MANUSCRIPT ORGANIZATION AND FORMATTING:

Reviewer #1 (Comments to the Authors (Required)):

ISG20 is a 3' exonuclease, belonging to interferon-stimulated genes, with broad antiviral activities. Its antiviral activity results from a complex mechanism including its capacity to specifically degrade viral RNA. However, some viruses can resist ISG20 exonuclease activity, indicating that viruses have evolved mechanisms to limit the ISG20 mediated degradation. The mechanisms governing cellular and viral mRNA resistance toward ISG20 are poorly characterized, and it has been recently demonstrated that RNA forming stable stem-loop structures or carrying 2'O methylation inside their own RNA are less sensitive to RNA degradation. One key question regarding the cellular expression of ISG20 is how cellular mRNA can resist ISG20, especially when its expression is increased in interferon-stimulated cells. This work addresses the important question of how mRNA are protected from ISG20 mediated degradation in cells transduced with ISG20 expression constructs. The authors first deployed an "unbiased" approach based on RNAseq performed on HEK293T cells expressing or not ISG20. This approach reveals that there are no drastic changes in the RNA transcriptome except the decrease of a small subset of histone related RNA known to lack polyadenylation but harboring 3' terminal stem loops. The authors next identified some elements participating in the resistance of RNA toward ISG20 degradation. The most interesting observation is that the Poly A binding protein (PABP1) can be an active player in mRNA protection. Indeed, they show through in vitro experiments that the binding of PABP1 protects synthetic poly A against ISG20 degradation. Even though this is an important observation, the present work did not provide direct evidence demonstrating that this process is a key player in cells. In addition they provide data confirming the notion that the presence of stable stem-loop structures at the 3' end of RNA can also participate in ISG20 resistance of mRNA lacking poly A tail.

Thus overall the manuscript contains new interesting data explaining how mRNA can escape to ISG20 mediated degradation, but need further work to demonstrate the role of PABP1 in a cellular context.

major comments

A table showing the 208 and 230 genes up and down regulated after ISG20 expression must be included in the supplementary section and the effect of M1 ISG20 on those genes should be included to ensure that the exonuclease activity is a key player for RNA expression level control. This is especially important as we have no information about the endogenous ISG20 expression level in HEK cells. In addition, how ISG20 can induce the upregulation of gene expression is puzzling as it is not expected for an RNase. This should be discussed in the manuscript.

The authors state that the presence of a stable stem loop structure present at the 3' end of histone related RNA can prevent ISG20 mediated degradation. However the specific impact of this stem loop structure is not shown in the manuscript. Rather, this hypothesis was tested by an in vitro assay performed on different RNAs corresponding to the HIV TAR structure and in cells infected with MOPV. This should be clarified in text and additional biochemical experiments are needed.

The paragraph describing the RNA protection by PABP1 is the key point of this manuscript and should be moved before the analysis of stem loop protection. The biochemical exonuclease assay needs the following additional control experiments: 1) include a control RNA lacking the 20 bases poly A tail (in figure 4B and C), 2) perform dose response inhibition of the ISG20 exonuclease activity by PABP1, 3) test the protection of PABP1 on mRNA containing a poly A tail extracted from cells and on cellular RNA devoid of poly A tail.

The protection ensured by PABP1 needs to be confirmed in cells, as these mRNAs are quite different from synthetic ones due to the high level of post-transcriptional modifications, as described in the introduction of the manuscript. Otherwise, the conclusions of the paper should take into account that the role of PABP1 in cells is not formally demonstrated by this study.

Minor comments

Abstract:

Add a sentence describing the result of the RNASeq which is key and new.

The results indicated that PABP1 can protect the poly A tail from ISG20-mediated degradation in vitro, however the direct role of PABP1 RNA protection was not demonstrated by experiments performed on cellular RNA or in cells. The authors should avoid over interpretation of their results in the abstract.

The conclusion of the abstract should be less speculative because the manuscript does not focus on the role of PABP1 on viral RNA.

Introduction:

The acronym ISG corresponds to "interferon stimulated gene" and not to "interferon sensitive gene". Please change the end of last sentence of the first paragraph "to be fully appreciated" by "remains to be unraveled". The epitranscriptomic RNA modifications are not only co-transcriptional modifications as stated in the introduction. ISG20 has no DNase activity even though it was proposed in early papers. The third sentence of the third paragraph is confusing as 2'O-methylation cannot be removed by eraser. To my knowledge it is mainly the N6 methylation that can be erased by specific cellular proteins. The authors claim that "The fact that epitranscriptomic modifications can modulate the susceptibility of RNAs to the action of ISG20 opens up to the unaddressed question of how this enzyme behaves towards cellular RNAs in which these modifications are largely present. To more globally appreciate the effects of ISG20 on RNAs at the scale of the whole cells, we performed an RNAseq analysis on cellular mRNAs." However the role of such modification in the RNA protection is not addressed in the Manuscript that focuses on the role of PABP1 and 3' end stem loop structure on RNA protection against ISG20 exonuclease. A paragraph describing PABP1 is needed in the introduction. The HIV genome length is about \approx 10 kB and not 9 kB. The end of the introduction focuses on viral RNA, whereas most results presented in the manuscript address the question of how cellular mRNAs are protected from ISG20 degradation. Please reformulate

Results:

For the second sentence, please add at the beginning of the sentence : "The expression level of
The antibodies used for WB analysis must be described in the figure legend.

The basal expression of ISG20 in HEK cells is a key point for the interpretation of the RNA seq analysis and is not discussed in the manuscript (RT-QPCR, or WB using anti-ISG20 antibodies).
The ref 17 does not correspond to a paper describing the effect of ISG20 mutation on its enzymatic activity.
The RNA seq data are quite important but are barely explained and should be more carefully described.
Please indicate that the RNAseq enrichment data could be linked to the exonuclease activity of ISG20 as mutated ISG20 was included in this study
Figure 1F: the colour code of WT ISG20 should be orange
The overall organization of the manuscript is puzzling as the RNA seq shows that the protection is mostly dependent on the poly A tail and to a lesser extent by the 3' stem-loop structure present in histone mRNA. I would suggest presenting first the biochemical data obtained with PABP1 protection regarding ISG20 activity, and next to focus on the role of 3' end stem-loop. In addition the role of histone 3' stem-loop is not addressed by biochemical assay.
In the text presenting the WB of figure 2B, it should be mentioned that the viral protein analyzed by WB is the Z protein and the result of the WB is not explained in the result section.
The authors assume that the protection effect observed on MOPV RNA is representative of mRNA protection of histone related RNA observed by RNA seq. This assertion is based on the absence of any effect of ISG20 on MOPV RNA and protein expression in A549 cell infection and must be discussed. In addition there is no negative control experiment showing that a virus and/or an RNA without such a stem-loop structure is sensitive to ISG20 mediated degradation in A549 cells. The escape of MOPV to ISG20 restriction should also be further discussed regarding the ISG20 subcellular localization, as the compartmentalization of the viral replication machinery in membrane vesicles and the possibility that the RNA protection can participate in viral RNA protection toward ISG20 exonuclease.
The paragraph describing the distributive activity of ISG20 is not key for this study as 1) the distributive activity of ISG20 is already assessed by the laddering RNA degradation pattern observed and discussed in several previous papers 2) is not an important point regarding the RNA protection by PABP1 or by the presence of stable hairpin structure at the 3' end of RNA. I would suggest merging the description of the biochemical assay with the first part of the paragraph describing the in vitro assay performed on stem-loops or poly A.
Figure 3 or 4: a catalytic mutant of ISG20 must be included in the experiments.
The biochemical data presented in the last paragraph concern well characterized TAR stem-loops of HIV-1, located at the 3' end of viral RNA. Although the rationale of this assay is explained, it is puzzling as the protection described in cells concern other stem-loops present at the 3' of histone related proteins or in the genome of MOPV. I suggest including such stem loops in the enzymatic assay presented in the figure 5.

Discussion

The overall interpretation regarding the conclusion must be toned down since the interesting protection of RNA by PABP1 is not yet directly proven in cells. Although the in vitro data demonstrates that PABP1 binds to synthetic poly A and shields them from ISG20 mediated degradation, the result combined with the RNA seq data are not sufficient to support the hypothesis that PABP1 is the key element of mRNA protection in cells.

Reviewer #2 (Comments to the Authors (Required)):

ISG20 is an interferon stimulated gene that inhibits viral replication through at least two proposed mechanisms: targeted RNA degradation via its 3' RNA exonuclease activity and inhibiting mRNA translation. In this manuscript, Louvat et al first use RNA-seq to identify which cellular mRNAs are degraded by overexpressed ISG20 in HEK293T cells. They found that the abundance

of most mRNAs is not affected by ISG20 expression and there is only a small effect for the ones that are, which is consistent with other reports that have overexpressed ISG20. There was an enrichment of histone mRNAs in the group of mRNAs that were downregulated by ISG20 expression. The authors then analyzed why most cellular mRNAs are not targeted by ISG20. They show that RNAs can be protected through two mechanisms. First, if the RNA has a poly(A) tail, PABP1 binding protects the RNA from degradation. Second, if the RNA has a terminal 3' stem-loop structure with high enough stability, the RNA will be protected. These results are what one would expect because ISG20 is a 3' exonuclease and the role of 3' RNA structure has been explored in other papers as well. The authors discuss how RNA modifications, such as m6A, may regulate ISG20 activity but the data in this manuscript does not impact upon that. The role of ISG20 for inhibiting translation, which may be more important for its antiviral activity, is not explored in this paper, though this has been analyzed by the Cimarelli lab previously. Overall, the implication of this paper is that most cellular mRNAs are protected from ISG20-mediated degradation and many viral mRNAs would also be protected from this exonuclease through the same mechanisms.

Major comments:

1. The authors discuss how ISG20 acts as a distributive enzyme but this is not well defined and it is not clear how the experiments in Figure 3 clearly show this. It would be good to expand on this idea so that more readers will be able to follow their logic.
2. Do the authors think that ISG20 expression in the context of physiological levels of interferon affect histone mRNA abundance and therefore DNA structure and gene expression? This is not clear in the manuscript.
3. In the discussion (page 14, last sentence in the first paragraph), the authors say that a large number of viruses do not have a poly(A) tail. I do not think this is correct, many human RNA viruses do have a poly(A) tail on their mRNAs that can be produced using several different mechanisms. Therefore, most viral mRNAs should be resistant to ISG20-mediated decay because PABP1 will bind the tail, though the authors could potentially expand their argument to differentiate between viral genomic RNAs and mRNAs. There is no evidence that I am aware of that PABP1 would not bind viral poly(A) tails. This means that the presence of a poly(A) tail is unlikely to be the mechanism by which ISG20 could differentiate between cellular and viral mRNA in the context of most viral infections of human cells. Viral RNAs that don't have a poly(A) tail likely have a structured 3' end, which will also protect the viral RNA from ISG20, or be bound to viral nucleoproteins. The authors should be clearer about the implications of this work on whether the major antiviral activity of ISG20 is to target viral RNA for degradation or whether it has another mechanism, such as inhibiting translation.

Please, find below our answers (blue) to the concerns expressed by the reviewers (black).

REVIEWER 1**GENERAL COMMENTS**

The reviewer defines our study as *containing new interesting data explaining how mRNA can escape ISG20-mediated degradation*, but suggests a few experiments to ameliorate our study.

Accordingly, we have now introduced several experiments and analyses (Figs 3C, 3D, 4, 6, SuppFig 1A, SuppFig 3 and SuppTable 3) that collectively support the conclusions of our study.

MAJOR CONCERNS

1) The reviewer asks to include a table of the genes modulated in the presence of WT ISG20, to compare them to those observed with the M1 ISG20 mutant and finally to discuss how ISG20 could induce such transcriptional changes, which is unexpected for an RNase.

Accordingly, we have listed all the genes modulated by WT ISG20 and also shown the corresponding changes obtained in the presence of the catalytically-inactive M1 ISG20 mutant (Supplementary Table 3). Our hypothesis is that the decrease in histone mRNAs may itself drive subtle changes in chromatin reorganization, leading to transcriptomic changes. We have put forward this hypothesis in the Discussion section (page 14).

2) The reviewer indicates that the impact of the histone and Mopeia virus RNAs stem-loops has not been tested specifically *in vitro*, given that our *in vitro* assays used stem-loops derived from an HIV structure (TAR).

We agree with the reviewer and we have now carried out such experiments with both histone and MOPV stem-loop *in vitro* (Fig 6). Experiments with the TAR structure (Fig 7) were meant to define more broadly the link between structure stability and susceptibility to ISG20. These figures are presented separately, as we have not been able to combine them without loss of definition of the characters and images.

3-4) The reviewer suggests to move experiments describing the role of PABP1 before the stem-loop part and also suggests a few experiments, in particular: include a control RNA lacking the poly(A) tail; perform a dose response inhibition of ISG20 by PABP1; test the protection of PABP1 on poly(A) mRNAs extracted from cells and more generally provide further evidence of the protective role of PABP1.

According to the suggestions of the reviewer, we have moved the PABP1 part prior to the stem loops; we have performed PABP1 dose-response experiments (Fig 3C and 3D); added a control experiment on a non-poly(A) mRNA in the presence of PABP1 (Supp Fig 3C) and performed experiments on cellular mRNAs in support of a protective role of PABP1 against ISG20 (Fig 4). In this case, we have decided not to carry out silencing experiments of PABP1, which by its pleiotropic roles would have made a direct interpretation of specific effects of

ISG20 very difficult. Instead, we have isolated cellular poly(A) mRNAs either without associated proteins, or in complex with PABP1, following immunoprecipitation with an anti-PABP1 antibody and we have then determined their susceptibility to ISG20 *in vitro* (Fig4). Our results indicate that PABP1-bound cellular mRNAs are indeed protected from ISG20.

MINOR CONCERNS

Abstract

- 1) We have added a sentence in the Abstract section, describing the RNAseq data as suggested.
- 2) We have added additional experiments to strengthen our conclusions that PABP1 protects cellular mRNAs from ISG20 (Fig 4).
- 3) We have rephrased the last sentence of our abstract.

Introduction

- 4) We have replaced *sensitive* with *stimulated*.
- 5) We have changed the last sentence of the first paragraph to *remains to be unraveled*.
- 6) We have rephrased the sentence on epitranscriptomic RNA modifications not to give the idea that they are only co-transcriptional.
- 7) The DNase activity refers to members of the DEDD superfamily, not to ISG20, which is defined as early as the first line as an RNase.
- 8) We have rephrased the sentence so as not to give the impression that the 2'O-methylation can be reversed, as suggested.
- 9) We agree with the reviewer that our study did not address directly the role of epitranscriptomic modifications on the susceptibility of cellular RNAs to ISG20. However, the fact remains that such modifications are admitted to be largely present in cellular mRNAs and this constituted the starting point of our study. We have specified this point (page 14).
- 10) We have introduced PABP1 (first paragraph page 4).
- 11) We have corrected the size of the HIV genome to $\approx 10kb$
- 12) We have reformulated the end of the introduction

Results

- 13) We rephrased the second sentence as suggested.
- 14) We have specified the antibodies used for WB in the Legends of each Figure.
- 15) We have already published on the levels of endogenous ISG20 expression (PPath-2019 Supp Fig1; <https://journals.plos.org/plospathogens/article?id=10.1371/journal.ppat.1008093#sec028>). In our hands, HEK293T cells do not express detectable levels of ISG20, as assessed by RT-qPCR. We confirm that we have not identified a reliable source of antibody capable of recognizing the endogenous form of ISG20.

16) Ref 17 (Wu et al 2019) refers to our publication in which the RNase activity of the ISG20 M1 mutant is clearly assessed (Supp Fig 7C; <https://journals.plos.org/plospathogens/article?id=10.1371/journal.ppat.1008093#sec028>). 17-18) We have better commented on the RNAseq results and the link of changes to the enzymatic activity of ISG20 (page 10, 14, novel Supplementary Figure 1A and Supplementary Table 3).

19) We have changed the colors of Fig 1F, as suggested.

20) According to the suggestion of the reviewer, we have moved the PABP1 part prior to the stem-loop one.

21) We have mentioned the Z protein in the result section (page 12), corresponding to Fig 5B (previously Fig 2B).

22) We have directly examined the role of the MOPV stem-loop in vitro (Fig 6). Unfortunately, this region is key to the viral genome replication and as such, mutations that disrupt the stem-loop result inevitably in replicative-defective viruses.

23) We have commented on the possibility that resistance/susceptibility of viral RNAs to ISG20 could in general be modulated also by the compartmentalization of the viral replication machinery (page 13).

24) We believe the paragraph on the distributive nature of the activity of ISG20 is important not only for its connections with PABP1, but more generally for its connections to other RNA binding factors with which ISG20 is likely to confront itself. Laddering is not per se an absolute proof of distributivity as a processive exonuclease can bind the RNA and digest one nucleotide at a time, while remaining bound to the substrate. This is the case for processive 3'- 5' exonucleases such as WRN, SsoXPF and TREX1 (Machwe et al. 2006, doi:10.1186/1471-2199-7-6; Roberts and White 2005, doi:10.1093/nar/gki974). Yuan et al. 2015, DOI:10.1074/jbc.M115.653915).

25) We have examined the catalytically inactive mutant M1 (Supp Fig 3A).

26) We have used the well-characterized TAR structure to provide a broader perspective on how the thermodynamic stability of stem-loops may influence the susceptibility of target RNAs to ISG20. However, we also agree with the reviewer that the weight of the specific stem loops derived from histone mRNAs and MOPV genome should have been examined directly. We have now performed such experiments (Fig 6).

Discussion

27) In keeping with the remarks of Reviewer 2, we have restructured the entire Discussion.

REVIEWER 2

GENERAL COMMENTS

The reviewer appears positive about our study and raises a few comments that we address below.

1) In the general statement, the reviewer indicates that the fact that PABP1, or stem-loops would protect RNAs from ISG20 is expected and has been explored in other papers, as well.

We disagree with the reviewer: no study examined the consequences that PABP1, or for that matter other RNA binding proteins, may bear on the biochemical activity of ISG20. While it is true that certain studies analyzed the role of specific stem-loops in the susceptibility to ISG20, this is the first study that does it systematically by linking thermodynamic stability of the 3' end structure to ISG20 susceptibility, which is in our view a key parameter not examined in prior studies. As such, we believe our findings are novel.

2) The reviewer comments that while we discuss about epitranscriptomic modifications, our study does not directly examine their weight in the susceptibility of RNAs to ISG20.

We agree with the reviewer and our study does not focus on the weight that epitranscriptomic modifications may have on ISG20 activity. Indeed, such modifications are mentioned only in the introduction and in the discussion sections to provide the reader with a comprehensive overview of the elements that have been described to modulate the susceptibility of target RNAs to ISG20. Within this context, our study introduces two novel elements of RNA protection against ISG20. We have rephrased certain passages to clarify this topic.

3) The reviewer indicates that we have not explored other mechanisms of ISG20 inhibition, notably the one of translation inhibition that our lab has reported.

The reviewer is correct and in this study we try to identify elements that protect RNA from ISG20. We believe this is important irrespectively of the mechanism used by ISG20 to inhibit viral infection because of a very simple reason: in all mechanisms of inhibition described so far, the RNase ability of ISG20 is unquestioned. In this respect, the proposed mechanisms of viral inhibition by ISG20 differ only for the nature of the RNA that is degraded: viral RNA in the model of direct viral RNA degradation and a cellular RNA to be identified in the model of translation inhibition). The identification of elements that generally provide protection from ISG20 is therefore important for one or the other mechanisms. We have restructured the entire Discussion to include this point.

MAJOR ISSUES

1) The reviewer asks us to clarify to the reader the experiment on distributivity.

We agree with the reviewer and we have now better explained this topic with a more complete scheme (Fig 2A) and text description (page 11).

2) The reviewer asks us to discuss whether the changes we have observed on the histone mRNA abundance may be observed in the context of IFN responses.

We ignore the answer to this interesting question and we have raised it in the Discussion section (page 14).

2) The reviewer asks us for a more extensive discussion about the implications of our findings for the current two models of ISG20-mediated viral inhibition (namely, direct viral RNA degradation or inhibition of viral RNA translation).

We agree with the reviewer and we have restructured the entire Discussion section to include this point.

MINOR ISSUES

None expressed

February 22, 2024

RE: Life Science Alliance Manuscript #LSA-2023-02233-TR

Dr. Andrea Cimarelli
International Center for Infectiology Research
46 Allée d'Italie
Lyon 69007
France

Dear Dr. Cimarelli,

Thank you for submitting your revised manuscript entitled "Stable Structures or PABP1 loading protect cellular and viral RNAs against ISG20-mediated decay". We would be happy to publish your paper in Life Science Alliance pending final revisions necessary to meet our formatting guidelines.

- please address Reviewer 1's remaining comments
- please be sure that the authorship listing and order is correct
- Please upload only clean version of manuscript file in .docx file format as main manuscript file
- Please upload all figure files as individual ones, including the supplementary figure files; all figure legends should only appear in the main manuscript file ('Figure legends" and 'Supplementary figure legends')
- Please upload each supplementary table separately in .docx or excel file format. Table legends should be placed after the figure legends in main manuscript file.
- Title in manuscript and the system need to match
- Please move 'Materials and methods' section after the 'Discussion' section
- Please add a callout for Figure 4A and 4B to your main manuscript text
- Please add an Author Contributions section to your main manuscript text

A. FINAL FILES:

B. MANUSCRIPT ORGANIZATION AND FORMATTING:

Sincerely,

Reviewer #1 (Comments to the Authors (Required)):

revision 1 of the manuscript "Stable Structures or PABP1 loading protect cellular and viral RNAs against ISG20-mediated decay"

ISG20 is a 3' exonuclease with broad antiviral activities. Its antiviral efficacy results from a complex mechanism, including its capacity to specifically degrade viral RNA and to limit RNA translation into viral proteins. However, some viruses can resist ISG20 exonuclease activity, indicating that viruses have evolved mechanisms to limit ISG20 antiviral activity. The mechanisms governing cellular and viral mRNA resistance toward ISG20 are poorly characterized, and it has been recently demonstrated that RNA forming stable stem-loop structures or carrying 2'O methylation are less sensitive to RNA degradation. One key question regarding the cellular expression of ISG20 is how ISG20 discriminates self from non-self RNA and how cellular mRNA can resist ISG20

. This work addresses this important question by deploying an agnostic approach based on RNAseq performed on HEK293T cells expressing or not expressing ISG20. This approach reveals that there are no drastic changes in the RNA transcriptome except for the decrease of a small subset of histone-related RNA known to lack polyadenylation but harbor 3' terminal stem loops. The authors next identified some elements participating in the resistance of RNA toward ISG20-mediated degradation. The authors postulated that the Poly A binding protein (PABP1) can be an active player in mRNA protection. Accordingly, they show through in vitro experiments that the binding of PABP1 protects synthetic poly A against ISG20 degradation, and the protection was confirmed by extracting cellular RNA using native conditions. In addition, they also provide data confirming the notion that the presence of stable stem-loop structures at the 3' end of RNA can also participate in ISG20 resistance of mRNA lacking a poly A tail. The RNA protection seems to depend on the stability of the 3' end hairpin structure. Thus, overall, the manuscript contains new interesting data explaining how self and viral RNA can escape ISG20-mediated degradation, and the manuscript can be published after some minor revisions (see below).

Minor Comments:

Abstract:

I would suggest reformulating the following sentences:

"Terminal stem loop RNA structures, that have been associated with ISG20 protection before but that we re-examine here systematically, can also provide protection against ISG20 but in this case, the balance between resistance and susceptibility to ISG20 depends on their thermodynamic stability."

to:

"Terminal stem loop RNA structures have been previously associated with ISG20 protection. We re-examine how these structures provide protection against ISG20, and we observed that the balance between resistance and susceptibility to ISG20 depends on their thermodynamic stability."

Introduction:

Line 60: Even though it has been shown in early papers, ISG20 has no DNase activity.: Please delete "or DNase".

Line 103: Please add ")" after ref 25.

Line 107: Please change "against ISG20." to "against ISG20-mediated degradation".

Line 411: I would suggest adding at the end of the sentence "exonuclease activity."

Line 421: Please correct the typo of "in vitro" to "in vitro".

Line 423: Please note the "(representative agarose gel panels and their densitometric quantification)" in the legend of the figure.

Line 428: Please add an introductory sentence such as "As the agnostic analysis of cellular RNA stability indicates that histone RNAs with a 3' stem-loop structure are more sensitive to ISG20-mediated degradation than polyadenylated RNAs, we next dissected the level of RNA protection ensured by 3' end stem-loop structures".

Referee Cross-Comments' to your review report.

no specific comments.

Reviewer #2 (Comments to the Authors (Required)):

The authors have revised the manuscript and improved it. My comments on the manuscript have been addressed.

February 23, 2024

RE: Life Science Alliance Manuscript #LSA-2023-02233-TRR

Dr. Andrea Cimarelli
International Center for Infectiology Research
46 Allée d'Italie
Lyon 69007
France

Dear Dr. Cimarelli,

Thank you for submitting your Research Article entitled "Stable Structures or PABP1 loading protect cellular and viral RNAs against ISG20-mediated decay". It is a pleasure to let you know that your manuscript is now accepted for publication in Life Science Alliance. Congratulations on this interesting work.

DISTRIBUTION OF MATERIALS:

Again, congratulations on a very nice paper. I hope you found the review process to be constructive and are pleased with how the manuscript was handled editorially. We look forward to future exciting submissions from your lab.

Sincerely,
